# Determinants of Adverse Perinatal Outcomes in Ibadan, Nigeria: The influence of maternal lifestyle

Ikeola A. Adeoye[1,2*], Chioma O. Unogu[1], Kofoworola Adediran[3], Babatunde M. Gbadebo[1]

1 Department of Epidemiology and Medical Statistics, Faculty of Public Health, College of Medicine, University of Ibadan, Ibadan, Nigeria, 2 Consortium of Advanced Research Training in Africa, Nairobi, Kenya, 3 Institute of Child Health, College of Medicine, University of Ibadan, Ibadan, Nigeria

* adeoyeikeola@yahoo.com, iadeoye@cartafrica.org

## Abstract

Adverse perinatal outcomes (APO) are unfavourable incidents of at least one of the following: low birth weight, preterm delivery, stillbirths, neonatal deaths, and perinatal deaths. They contribute significantly to neonatal and infant morbidity and mortality, developmental abnormalities, and long-term impairments. Studies are lacking on the influence of maternal lifestyle on APO in Nigeria. Hence, we investigated the determinants of APO using the Ibadan Pregnancy Cohort Study (IbPCS) data and examined 1,339 mothers who had hospital delivery. The outcome variable was APO (low birth weight, birth asphyxia and preterm delivery). Explanatory variables comprised Antepartum Depression (Edinburgh Depression Scale ≥ 12), Physical activity (Pregnancy Physical Activity Questionnaire (PPAQ), Dietary pattern (Qualitative Food Frequency Questionnaire (FFQ), Maternal Stress (Perceived Stress Scale), Alcohol consumption, and Tobacco exposure. We used binary and multiple logistic regression to assess the associations between the risk factors and adverse perinatal outcomes at a significant P-value <0.05. Prevalence of APO was 26.7%, 95%CI (24.4–29.1); low birth weight - 8.5%, 95%CI (7.0–10.1) preterm delivery 14.8%, 95% CI (12.9–16.7); birth asphyxia 16.3%, 95%CI (14.0–18.9). The factors associated with LBW were being a female infant AOR: 2.00, 95%CI (1.13 -3.52); emergency caesarean section AOR: 2.40, 95%CI (1.06–5.42); a history of hypertension in pregnancy AOR: 3.34, 95%CI (1.45 -7.52). Preterm birth was associated with being poor AOR: 2.00, 95%CI (1.13 -3.52); history of stillbirth AOR: 2.05, 95%CI (1.14–3.68); antepartum depression AOR: 1.87, 95%CI (1.08–3.25). Of the lifestyle factors examined, only a high protein diet with a non-alcoholic beverage dietary pattern had a statistically significant association with preterm birth [AOR: 0.50, 95%CI (1.08–3.52)]. However, lifestyle factors had no significant association with LBW and birth asphyxia in our study. Understanding these risk factors can help policymakers and healthcare professionals create cost-effective interventions to curtail the burden of APO in Nigeria.

**Data availability statement:** The Ibadan Pregnancy Cohort Study datasets generated and analysed during the current study are not publicly available because they contain potentially identifying and confdential information but are available from the UI/UCH Ethics Committee (uiuchec@gmail.com) on reasonable request if it meets the criteria for accessing confdential data.

**Funding:** This research was supported by the Consortium for Advanced Research Training in Africa (CARTA). CARTA is jointly led by the African Population and Health Research Center and the University of the Witwatersrand and funded by the Carnegie Corporation of New York (Grant No–B 8606.R02), Sida (Grant No:54100113), the DELTAS Africa Initiative (Grant No: 107768/Z/15/Z) and Deutscher Akademischer Austauschdienst (DAAD). The DELTAS Africa Initiative is an independent funding scheme of the African Academy of Sciences (AAS) 's Alliance for Accelerating Excellence in Science in Africa (AESA) and supported by the New Partnership for Africa's Development Planning and Coordinating Agency (NEPAD) with funding from the Wellcome Trust (UK) and the UK government. Ikeola Adeoye is a CARTA PhD fellow. The statements made and views expressed are solely the responsibility of the Fellow. The funders had no role in study design, data collection and analysis, decision to publish, or preparation of the manuscript.

**Competing interests:** The authors have declared that no competing interests exist.

## Introduction

The World Health Organization (2019) reported that 75% of newborn deaths occur in the first week of life, and 1 million of these deaths occur within the first 24 hours, with infections, preterm births and birth asphyxia being the leading causes of death [1]. Increased efforts to meet the Millennium Development Goals in the past and now the Sustainable Development Goals (SDG) led to a global decrease in the perinatal mortality rate from 37 deaths per 1000 live births in 1990 to 18 deaths per 1000 live births in 2021 [2]. Adverse perinatal outcomes (APOs) refer to unfavourable outcomes during pregnancy, childbirth, and the early neonatal period, and these include low birth weight (birth weight < 2,500 g), low Apgar score at the first or fifth minutes after delivery, preterm delivery (before 37 weeks of gestation), stillbirth (fetal deaths occurring after the age of viability), gross congenital anomaly or neonatal death within 24 hours and admission to the neonatal intensive care unit either during pregnancy, immediately after birth and within seven days after birth [3–5].

APOs are the leading causes of neonatal and infant mortality worldwide, and ending preventable neonatal death has been a significant global public health concern [6,7]. They are an indicator of quality intrapartum and early postpartum healthcare. APOs are substantial determinants of perinatal survival, infant morbidity and mortality and long-term disability such as neurological and developmental disabilities. For instance, babies born with low birth weight have an increased risk of future non-communicable diseases. Adverse perinatal outcomes (APOs) indicate the quality of perinatal, intrapartum, and early postpartum healthcare [8]. Notably, APOs have drastically declined in high-income countries (HIC), but they remain high in low- and middle-income countries (LMIC), varying across countries [9]. While 1 in every 300 neonates dies in HIC, 1 in 36 die in sub-Saharan Africa (sSA) [10]. Moreover, West Africa reports the highest perinatal mortality rate of 65.1 per 1000 total births compared to 49.9 per 1000 total births and 56 per 1000 total births in East Africa and South Africa, respectively [9].

Even though there is inadequate evidence on APOs in sSA, a broad range of factors have been implicated – sociodemographic, obstetrics, nutritional, and health systems issues [11–14]. A recent study [15] examined the determinants of APOs in sSA using the most recent Demographic Health Surveys in ten sSA countries reported maternal education, socioeconomic status, intimate partner violence, the gender of the unborn child, twin gestation, women's autonomy for health care decision-making, delay in accessing perinatal care, quality of antenatal care (ANC) multi-parity, maternal anaemia, HIV [15]. Recently, attention has been drawn to the role of maternal lifestyle in perinatal health and outcomes, but there have been limited studies in sub-Saharan Africa, including Nigeria [16]. A mother's lifestyle, including her mental health status, such as diet, exercise, smoking, drug use, stress, anxiety, and melancholy, are critical modifiable determinants of adverse perinatal outcomes [17–19]. A study in China also revealed that physical activity contributed to an increase in the birth weight z-score and lowered the risk of preterm birth, supporting recommendations that pregnant women should be encouraged to engage in suitable physical activity during pregnancy [18].

Nigeria contributes to 6% of the global neonatal burden and has one of the highest perinatal mortality rates in Africa, with 76 per 1000 live births [20]. Research and programmatic efforts have been focused on perinatal deaths [20–22]. Still, there has been much less attention on APOs, which occur in more significant numbers in Nigeria, particularly the influence of modifiable maternal lifestyle factors. Therefore, we investigated APO's determinants, particularly maternal lifestyle factors, using the Ibadan Pregnancy Cohort Study in Nigeria.

## Materials and methods

### The Ibadan Pregnancy Cohort Study (IbPCS)

The Ibadan Pregnancy Cohort Study (IbPCS) has been documented elsewhere [23]. In brief, the IbPCS is a multi-centre, prospective cohort study conducted among antenatal care attendees at four major maternity hospitals within the Ibadan Metropolis, Oyo state, Ibadan. These are University College Hospital, Adeoyo Maternity Teaching Hospital, Jericho Specialist Hospital and Saint Mary Catholic Hospital Oluyoro Ibadan. The study was conducted from 1st April 2018–30th September 2019, and investigated the associations between maternal obesity, lifestyle factors, maternal metabolism (gestational hyperglycaemia, dyslipidemia, and gestational weight gain), pregnancy and postpartum outcomes. In all, 1745 pregnant women were recruited in early gestation (≤ 20 weeks gestation) and followed up until delivery. Data were collected using pretested, interviewer-administered questionnaires and desktop medical and nursing records review at different times after enrollment, third trimester and delivery of their babies. The eligibility criteria were: maternal age ≥ 18 years, early gestation ≤20 weeks. Women with severe medical complications were excluded from the study. The lifestyle factors examined included dietary patterns, sugar-sweetened beverages, physical activity, sedentary behaviour, tobacco use, alcohol consumption, and sleep patterns. We accessed the IbPCS data for research purposes from 15th June to 7th July 2023. The flow chart of study participants from recruitment until delivery by the occurrence of adverse perinatal outcomes is displayed in Fig 1.

## Operational definition

### Outcomes

Adverse perinatal outcomes (APOS) are the negative occurrence of at least one of the following: low birth weight, preterm delivery, stillbirths, neonatal deaths, and perinatal deaths. In this study, we assessed LBW, PTD and birth asphyxia. Low Birth Weight is birthweight < 2500 g; PTD < 37 completed weeks; birth asphyxia Apgar score < 7 at 1 minute [24].

### Lifestyle factors

**Dietary Pattern:** Five dietary patterns were extracted from the IbPCS study. A food frequency questionnaire evaluated 67 food and beverage intake among study participants before enrolment. The nutrition data was subjected to principal component analysis with varimax rotation. The dietary patterns were a protein-rich diet with non-alcoholic beverages and fruits and a typical diet with alcohol, legumes and refined grains. The details of the dietary patterns of IbPCS participants and the associated factors have been published [25].

**Physical Activity and Sedentary Behaviors Assessment:** The Pregnancy Physical Activity Questionnaire (PPAQ) evaluated physical activity during pregnancy. We assessed the intensity and types of physical activity across the various domains (household/care, occupational, transit and sport) and intensity (sedentary, mild, moderate and vigorous). The full details have also been documented elsewhere [26].

**Alcohol and Tobacco Exposure:** Alcohol intake and cigarette exposure were assessed by participants' self-reports of their presence or absence in answer to some questions [27].

**Unprescribed and Herbal Medicine:** The unprescribed and herbal medicine use was assessed in the index pregnancy's third trimester, and the types of drug & reason for drug use were obtained by self-report in answer to specific questions unique to both medicine use [28].

**Antepartum Depression/Perceived Stress:** Antepartum Depression was measured using the Edinburgh Postnatal Depression Scale (EPDS ≥12). The Perceived Stress Scale (PSS) also consists of 10 items. Each question contains four multiple-choice from which a total score

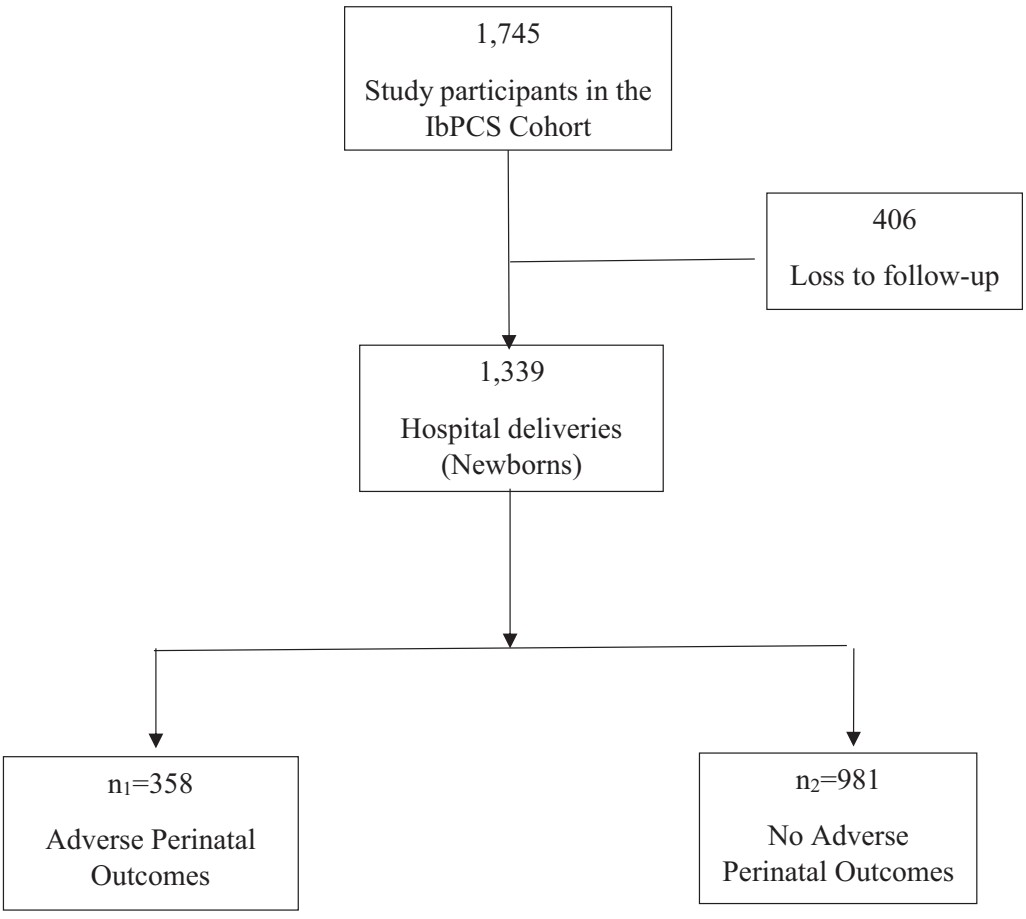

**Fig 1. Flowchart of study participants.**

of 40 is obtained and categorised into Low stress (0–13), Moderate stress (14–26), and High stress (27–40) [17].

**Body Mass Index (BMI):** These were categorised as underweight (18.5 kg/m2), normal weight (18.5–24.9 kg/m2), overweight (25.0–29.9 kg/m2), and obese (30 kg/m2) according to the WHO classification [29]. The neonatal measurements include weight (kg), length (cm), and head circumference (cm) of newborns extracted from medical records.

## Exposure variables

The sociodemographic variables included the following: maternal age (<35 years; ≥ 35 years), marital status (not married; married), woman's education (≤ primary, secondary, tertiary), employment status (unemployed; employed), religion (Christian; Muslim), monthly income in Naira (<20,000;20,000–99,999;≥ 100,000) and wealth tertiles (poor, middle, richest). Obstetric variables were history of stillbirth, history of gestational HBP, parity (nullipara, 1–3, and ≥4), and mode of delivery (spontaneous vertex delivery, elective cesarean section; the emergency cesarean section". Clinical variables - maternal BMI (underweight, normal weight, overweight, obese", antenatal visits (< 4visits; ≥ 4visits) and newborn gender. The lifestyles and behavioural variables included alcohol use, tobacco exposure, depression, unprescribed medicine, herbal medicines, and perceived stress.

## Data processing and analysis

Statistical analysis was performed using Stata version 13. Descriptive statistics summarized continuous data with means and standard deviation, and frequency and percentages for categorical data. The prevalence of APO and the specific causes – LBW, PTB and birth asphyxia were displayed graphically with bar graphs. The associations between respondent's background (Sociodemographic, maternal, obstetrical and medical) and lifestyle factors (like dietary patterns, physical activity, alcohol and smoking) and Adverse Perinatal Outcomes were tested using the chi-square test. Variables that were statistically significant at 5% level (history of stillbirth, mode of delivery, sex of the newborn, occupationally related physical activity) were subjected to multiple logistic regression analyses. Also the associations between respondent's background and lifestyle factors and the specific causes of APO - LBW, PTB and birth asphyxia examined using chi-square test (Table A and Table B in S1 Text), binary and multiple logistic regression analyses. Variables that were statistically significant at 5% level were subjected to multiple logistic regression analyses as follows LBW (sex of the newborn, mode of delivery, a history of gestational hypertension); PTB (maternal income, wealth index, mode of delivery, history of still birth, antepartum depression, sedentary activity, moderate intensity activity, household activity, protein rich diet and non- alcoholic beverages); birth asphyxia (parity, mode of delivery, history of miscarriage). Unadjusted and adjusted odds ratios and 95% confidence intervals (CI) and p-values were reported.

## Ethical consideration

The Oyo State Ministry of Health Ethical Committee (AD/13/479/710) and the University of Ibadan/ University College Hospital (UI/UCH) Institutional Review Board (UI/EC/15/0060) gave their approval for this study's ethical conduct. Before enrolling in the study, all respondents provided written and verbal informed consent. The Declaration of Helsinki, which stipulates confidentiality of data, beneficence toward participants, non-maleficence toward participants, and voluntariness, served as the basis for the study protocol and conduct. Minors (< 18 years) were not included in the study.

## Results

### Prevalence, sociodemographic, obstetric, and clinical characteristics of study participants by the APOS status (Table 1)

The background (sociodemographic, obstetric, clinical) characteristics of study participants by their APO status are shown in Table 1. The respondents' mean age and gestational age at delivery were 30.0 ± 5.2 years and 38.6 ± 2.1 weeks, respectively. The majority of the study participants were ≤ 35 years 1069 (79.8%), married 1261 (94.2%), and had ≥ four antenatal visits 583 (74.9%). The prevalence of APO was 26.7%. Specifically, APOS was commoner among older women ≥35 years 83 (30.6%), unmarried women 25 (32.1%), poorer women 124 (30.2%), nulliparous women 167 (28.2%), those with a history of hypertension in pregnancy 19 (36.5%), and neonates of the female sex 167 (28.2%). The prevalence of APO 26.7%, 95%CI (24.4–29.1); LBW 8.5% 95%CI (7.0–10.1); preterm 14.8%, 95%CI (12.9–16.7); birth asphyxia 16.3%, 95%CI (14.0–18.9) Fig 2.

### The lifestyle and behavioural characteristics of study participants (Table 2)

The lifestyle and behavioural characteristics of our study participants by their APOS status are shown in Table 2. In particular, alcohol use was 172 (12.9%), tobacco exposure was 48 (3.6%), and depression 224 (18.0%). However, with regards to the presence of APO, lifestyle factors

**Table 1. Background characteristics (sociodemographic, obstetric, and clinical) of the study participants by Adverse perinatal outcomes in Ibadan, Nigeria.**

| | Variables | Total | Adverse perinatal outcomes | | Chi-square | p-value |
|---|---|---|---|---|---|---|
| | | | Present | Absent | | |
| **OVERALL** | | **1339** | **26.7 (358)** | **73.3 (981)** | | |
| *Sociodemographic characteristics* | | | | | | |
| **Maternal age** | < 35 years | 1068 (79.76) | 275 (25.75) | 793 (74.25) | 2.626 | 0.105 |
| | ≥ 35 years | 271 (20.24) | 83 (30.63) | 188 (69.37) | | |
| **Marital status** | Not married | 78 (5.83) | 25 (32.05) | 53 (67.95) | 1.194 | 0.274 |
| | Married | 1261 (94.18) | 333 (26.41) | 928 (73.59) | | |
| **Woman's education** | ≤ Primary | 30 (2.25) | 8 (26.67) | 22 (73.33) | 2.469 | 0.291 |
| | Secondary | 346 (25.90) | 103 (29.77) | 243 (70.23) | | |
| | Tertiary | 960 (71.86) | 244 (25.42) | 716 (74.58) | | |
| **Employment status** | Unemployed | 156 (11.65) | 39 (25.00) | 117 (75.00) | 0.272 | 0.602 |
| | Employed | 1,183 (88.35) | 319 (26.97) | 864 (73.03) | | |
| **Religion** | Islam | 539 (40.50) | 136 (25.23) | 403 (74.77) | 0.960 | 0.327 |
| | Christianity | 792 (59.50) | 219 (27.65) | 573 (72.35) | | |
| **Monthly income** | <20,000 | 435 (37.12) | 125 (28.74) | 310 (71.26) | 3.632 | 0.163 |
| | 20,000 - 99,999 | 656 (55.97) | 159 (24.24) | 497 (75.76) | | |
| | ≥100,000 | 81 (6.91) | 25 (30.86) | 56 (69.14) | | |
| **Wealth tertiles** | Poorest | 410 (30.62) | 124 (30.24) | 286 (69.76) | 4.138 | 0.126 |
| | Middle | 451 (33.68) | 118 (26.16) | 333 (73.84) | | |
| | Richest | 478 (35.70) | 116 (24.27) | 362 (75.73) | | |
| *Obstetric characteristics* | | | | | | |
| **History of Contraceptive use** | Present | 208 (16.00) | 44 (21.15) | 164 (78.85) | 3.837 | 0.050 |
| | Absent | 1,112 (83.1) | 308 (27.70) | 804 (72.30) | | |
| **#History of Stillbirth** | Present | 119 (8.89) | 41 (34.45) | 78 (65.55) | 4.507 | **0.034***|
| | Absent | 769 (57.43) | 194 (25.23) | 575 (74.77) | | |
| **#History of Gestational HBP** | Present | 52 (3.88) | 19 (36.54) | 33 (63.46) | 3.554 | 0.059 |
| | Absent | 807 (60.27) | 200 (24.78) | 607 (75.22) | | |
| **Parity** | Nullipara | 593 (44.55) | 167 (28.16) | 426 (71.84) | 1.266 | 0.531 |
| | 1–3 | 678 (50.94) | 175 (25.81) | 503 (74.19) | | |
| | ≥ 4 | 60 (4.51) | 14 (23.33) | 46 (76.67) | | |
| **Mode of delivery*** | SVD | 814 (65.49) | 178 (21.87) | 636 (78.13) | 33.896 | **<0.001***|
| | Elective CS | 192 (15.45) | 73 (38.02) | 119 (61.98) | | |
| | Emergency CS | 237 (19.07) | 86 (36.29) | 151 (63.71) | | |
| | Others** | 79 (5.98) | 18 (22.78) | 61 (6.31) | | |
| *Clinical characteristics* | | | | | | |
| **Maternal BMI** | Underweight | 36 (2.77) | 8 (22.22) | 28 (77.78) | 0.759 | 0.859 |
| | Normal weight | 645 (49.62) | 177 (27.44) | 468 (72.56) | | |
| | Overweight | 373 (28.69) | 96 (25.74) | 277 (74.26) | | |
| | Obese | 246 (18.92) | 67 (27.24) | 179 (72.76) | | |
| **#Antenatal visits** | < 4 visits | 195 (25.06) | 48 (24.62) | 147 (75.38) | 0.162 | 0.687 |
| | ≥ 4 visits | 583 (74.94) | 152 (26.07) | 431 (73.93) | | |
| **Sex of newborn** | Male | 676 (52.94) | 158 (23.37) | 518 (76.63) | 5.147 | 0.023* |
| | Female | 601 (47.06) | 174 (28.95) | 427 (71.05) | | |

*SVD - Spontaneous vertex delivery; CS - cesarean section.

**Others - Induction of labour, Assisted breech delivery, Operative vaginal delivery.

\# 5% of data are missing.

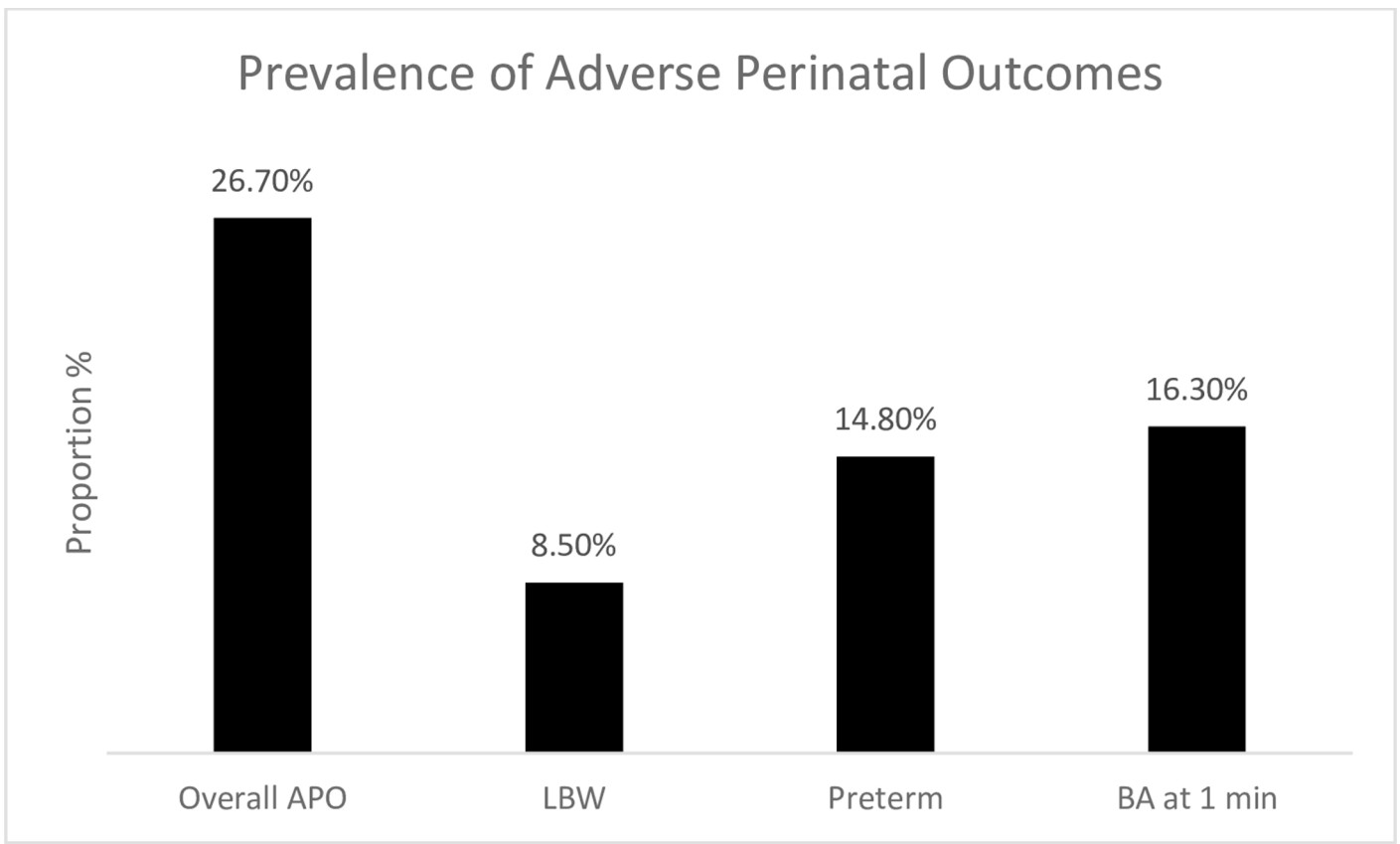

**Fig 2. Prevalence of adverse perinatal outcomes.**

were reportedly higher in women who had tobacco exposure, experienced depression during pregnancy 65 (29.0%) and used herbal drugs but lower in those with high levels of moderate intensity physical activity 101 (23.5%), high level of occupation-related activity 94 (22.1%) and high level of protein-rich diet with non-alcoholic beverage 98 (22.6%).

### Determinants of adverse perinatal outcomes (Tables 3–4)

The factors associated with Adverse Perinatal Outcomes in the Ibadan Pregnancy Cohort Study are displayed in Table 3. Women with highest tertile of occupation related activity had much lower odds for APO than the lowest tertile; adjusted OR [AOR]: - 0.59, 95% CI: 0.39–0.88; p = 0.010. On the other hand, a previous history of stillbirth [AOR]: - 1.59, 95% CI: 1.03–2.44; p = 0.036; emergency CS [AOR]: - 2.05, 95% CI: 1.36–3.10; p = 0.001, and elective CS [AOR]: - 2,34, 95% CI: 1.57–3,52; p = <0.001 increased the odds for APO compared with women with spontaneous vaginal delivery. The sex of the newborn became insignificant after adjustment for other variables.

### Determinants of specific adverse perinatal outcomes (Tables 4–5)

**Prevalence and associated factors of low birth weight.** The factors associated with LBW, PTB and birth asphyxia are shown in Tables 4-5. The prevalence of LBW was 8.5%. In the unadjusted model, factors associated with LBW were history of gestational hypertension [UOR=3.0; 95% CI:(1.42–3.43); p-value <0.004], birth by emergency caesarean section

**Table 2.  Lifestyle and behavioural characteristics of pregnant women by adverse perinatal outcomes in Ibadan, Nigeria.**

| Variables | | Total | Adverse Perinatal Outcomes | | Chi-square | p-value |
|---|---|---|---|---|---|---|
| | | (N=1339) | Present | Absent | | |
| **Alcohol use** | Yes | 172 (12.85) | 42 (24.42) | 130 (75.58) | 0.541 | 0.462 |
| | No | 1,167 (87.16) | 316 (27.08) | 851 (72.92) | | |
| **Tobacco exposure** | Yes | 48 (3.59) | 15 (31.25) | 33 (68.75) | 0.518 | 0.472 |
| | No | 1,291 (96.42) | 343 (26.57) | 948 (73.43) | | |
| **Perceived stress** | Low | 128 (10.65) | 39 (30.47) | 89 (69.53) | 2.100 | 0.350 |
| | Moderate | 999 (83.11) | 250 (25.03) | 749 (74.97) | | |
| | High | 75 (6.24) | 17 (22.67) | 58 (77.33) | | |
| **Depression** | Yes | 224 (17.96) | 65 (29.02) | 159 (70.98) | 1.612 | 0.204 |
| | No | 1,023 (82.04) | 255 (24.93) | 768 (75.07) | | |
| **Unprescribed medicines** | Yes | 159 (31.18) | 37 (23.27) | 122 (76.73) | 0.192 | 0.661 |
| | No | 351 (68.82) | 88 (25.07) | 263 (74.93) | | |
| **Herbal medicines** | Yes | 101 (21.49) | 31 (30.69) | 70 (69.31) | 1.230 | 0.267 |
| | No | 369 (78.51) | 93 (25.20) | 276 (74.80) | | |
| *Physical activity level Intensity* | | | | | | |
| **Sedentary intensity activity** | Low | 446 (33.31) | 128 (28.70) | 318 (71.30) | 1.413 | 0.493 |
| | Medium | 439 (32.79) | 111 (25.28) | 328 (74.72) | | |
| | High | 454 (32.91) | 119 (26.74) | 335 (73.79) | | |
| **Light intensity activity** | Low | 459 (34.28) | 131 (28.54) | 328 (71.46) | 1.439 | 0.487 |
| | Medium | 436 (32.56) | 109 (25.00) | 327 (75.00) | | |
| | High | 444 (33.16) | 118 (26.58) | 326 (73.42) | | |
| **Moderate intensity** | Low | 459 (34.28) | 126 (27.45) | 333 (72.55) | 3.731 | 0.155 |
| | Medium | 450 (33.61) | 131 (29.11) | 319 (70.89) | | |
| | High | 430 (32.11) | 101 (23.49) | 329 (76.51) | | |
| **Vigorous activity** | Low | 836 (62.44) | 219 (26.20) | 617 (73.80) | 1.218 | 0.544 |
| | Medium | 197 (14.71) | 59 (29.95) | 138 (70.05) | | |
| | High | 306 (22.85) | 80 (26.14) | 226 (73.86) | | |
| *Types* | | | | | | |
| **Occupation-related activity** | Low | 458 (34.20) | 135 (29.48) | 323 (70.52) | 6.945 | **0.031*** |
| | Medium | 456 (34.06) | 129 (28.29) | 327 (71.71) | | |
| | High | 425 (31.74) | 94 (22.12) | 331 (77.88) | | |
| **Transport-related activity** | Low | 511 (38.16) | 137 (26.81) | 374 (73.19) | 0.282 | 0.868 |
| | Medium | 407 (30.40) | 112 (27.52) | 295 (72.48) | | |
| | High | 421 (31.44) | 109 (25.89) | 312 (74.11) | | |
| **Household/caregiving activity** | Low | 453 (33.83) | 114 (25.17) | 339 (74.83) | 2.480 | 0.289 |
| | Medium | 448 (33.46) | 115 (25.67) | 333 (74.33) | | |
| | High | 438 (32.71) | 129 (29.45) | 309 (70.55) | | |
| **Sport or exercise activity** | Low | 490 (36.60) | 121 (24.69) | 369 (75.31) | 2.399 | 0.301 |
| | Medium | 432 (32.26) | 115 (26.62) | 317 (73.38) | | |
| | High | 417 (31.14) | 122 (29.26) | 295 (70.74) | | |
| *Dietary patterns* | | | | | | |
| **Protein-rich diet with non-alcoholic beverages** | Low | 449 (33.53) | 128 (28.51) | 321 (71.49) | 5.518 | 0.063 |
| | Medium | 457 (34.13) | 132 (28.88) | 325 (71.12) | | |
| | High | 433 (32.34) | 98 (22.63) | 335 (77.37) | | |
| **Fruits** | Low | 456 (34.06) | 113 (24.78) | 343 (75.22) | 5.917 | 0.051 |
| | Medium | 443 (33.08) | 137 (30.93) | 306 (69.07) | | |
| | High | 440 (32.86) | 108 (24.55) | 332 (75.45) | | |

*(Continued)*

**Table 2.** (Continued)

| Variables | | Total | Adverse Perinatal Outcomes | | Chi-square | p-value |
|---|---|---|---|---|---|---|
| | | (N=1339) | Present | Absent | | |
| **Typical diet with alcohol** | Low | 460 (34.35) | 116 (25.22) | 344 (74.78) | 2.067 | 0.356 |
| | Medium | 442 (33.01) | 129 (29.19) | 313 (70.81) | | |
| | High | 437 (32.64) | 113 (25.86) | 324 (74.14) | | |
| **Legumes** | Low | 472 (35.25) | 129 (27.33) | 343 (72.67) | 0.154 | 0.926 |
| | Medium | 458 (34.21) | 120 (26.20) | 338 (73.80) | | |
| | High | 409 (30.55) | 109 (26.65) | 300 (73.35) | | |
| **Refined grains** | Low | 446 (33.31) | 121 (27.13) | 325 (72.87) | 0.426 | 0.808 |
| | Medium | 452 (33.76) | 124 (27.43) | 328 (72.57) | | |
| | High | 441 (32.94) | 113 (25.62) | 328 (74.38) | | |

# ≥ #5% of data are missing.

**Table 3.** Factors associated with adverse perinatal outcomes (Adjusted Odds and 95% CI) in the ibadan pregnancy cohort study, Nigeria.

| Independent variables | Adjusted Odds Ratio | 95.0% Cl | | P value |
|---|---|---|---|---|
| **Occupation related activity** | | | | |
| Tertile 1 | 1 | | | |
| Tertile 2 | 0.79 | 0.55 | 1,15 | 0.220 |
| Tertile 3 | **0.59** | **0.39** | **0.88** | **0.010***  |
| Previous stillbirth | | | | |
| No | 1 | | | |
| Yes | **1.59** | **1.03** | **2.44** | **0.036***  |
| Mode of delivery | | | | |
| SVD | 1 | | | |
| EMCS | **2.05** | **1.36** | **3.10** | **0.001***  |
| ELCS | **2.34** | **1.57** | **3.52** | **<0.001** |
| Others | 1.44 | 0.69 | 2.98 | 0.330 |
| Sex of the newborn | | | | |
| Male | 1 | | | |
| Female | 1.16 | 0.84 | 1.59 | 0.367 |

[UOR=2.1; 95% CI:(1.32–3.29); p-value:0.002] and being a female newborn [UOR=1.9; 95% CI:(1.21–2.93); p-value=0.005] were statistically significant. Spontaneous vagina delivery had a negative association with LBW [UOR=0.56; 95% CI:(0.37–0.84); p-value: 0.005].

In the adjusted model, being a female infant [AOR=2.0; CI:(1.13–3.52); p-value: 0.017], emergency caesarean section (EMCS) [AOR= 2.4; CI:(1.06–5.42); p-value: 0.036] and history of gestational hypertension (HBP) [AOR= 3.3; CI: (1.48–7.52); p-value<0.004] remained significant. However, none of the lifestyle factors assessed was significantly associated with LBW.

**Prevalence and associated factors of the preterm birth.** The prevalence of PTB was 11.8% in this study. On univariate analysis, the significant determinants were income (UOR=0.70% CI: (0.48–0.99) p=0.043), wealth index (UOR=0.48, 95% CI: (0.32–0.71) p<0.001), history of stillbirth (UOR=1.84, 95% CI: (1.13–2.99) p=0.014), emergency caesarean section (UOR=1.63, 95% CI: (1.12–2.38) p=0.011), spontaneous vagina delivery UOR= 0.70, 95% CI: (0.51–0.96) p= 0.027), antepartum depression (UOR=1.73, 95% CI: (1.17–2.55)

**Table 4. Background factors (sociodemographic, obstetrical, clinical factors associated with adverse perinatal outcomes among pregnant women in Ibadan Nigeria.**

| Variables | Low Birth Weight | | | | Preterm Birth | | | | Birth Asphyxia | | | |
|---|---|---|---|---|---|---|---|---|---|---|---|---|
| Sociodemographic characteristics | Unadjusted | P-value | Adjusted | P-value | Unadjusted | P-value | Adjusted | P-value | Unadjusted | P-value | Adjusted | P-value |
| **Maternal age group** | | | | | | | | | | | | |
| <35years | 1 | - | - | - | 1 | - | - | - | 1 | - | - | - |
| ≥35years | 1.39 (0.87–2.23) | 0.168 | - | - | 1.11 (0.76–1.62) | 0.602 | - | - | 1.17 (0.75–1.82) | 0.497 | - | - |
| **Marital status** | | | | | | | | | | | | |
| Single | 1 | - | - | - | 1 | - | - | - | 1 | - | - | - |
| Married | 0.82 (0.36–1.84) | 0.625 | - | - | 0.56 (0.32–1.01) | 0.052 | - | - | 1.10 (0.51–2.39) | 0.803 | - | - |
| **Woman's education** | | | | | | | | | | | | |
| ≤Primary | 1 | - | - | - | 1 | - | - | - | 1 | - | - | - |
| Secondary | 0.85 (0.19–3.90) | 0.838 | - | - | 2.08 (0.61–7.10) | 0.245 | - | - | 0.62 (0.19–2.03) | 0.432 | - | - |
| Tertiary | 0.81 (0.18–3.55) | 0.777 | - | - | 1.25 (0.37–4.20) | 0.720 | - | - | 0.63 (0.20–1.95) | 0.419 | - | - |
| **Employment status** | | | | | | | | | | | | |
| Unemployed | 1 | - | - | - | 1 | - | - | - | 1 | - | - | - |
| Employed | 0.84 (0.47–1.52) | 0.569 | - | - | 0.20 (0.13–0.31) | 0.472 | - | - | 0.81 (0.94–3.47) | 0.075 | - | - |
| **Religion** | | | | | | | | | | | | |
| Islam | 0.79 (0.51–1.22) | 0.281 | - | - | 1.06 (0.77–1.46) | 0.716 | - | - | 0.76 (0.52–1.2) | 0.166 | - | - |
| Christianity | 1 | - | - | - | 1 | - | - | - | 1 | - | - | - |
| **Ethnicity** | | | | | | | | | | | | |
| Non-Yoruba | 1 | - | - | - | 1 | - | - | - | 1 | - | - | - |
| Yoruba | 1.08 (0.56–2.08) | 0.811 | - | - | 1.13 (0.67–.90) | 0.657 | - | - | 0.85 (0.50–1.45) | 0.545 | - | - |
| **Maternal Income** | | | | | | | | | | | | |
| <20,000 | 1 | - | - | - | 1 | - | 1 | - | 1 | - | - | - |
| 20,000 - 99,999 | 0.76 (0.48–1.21) | 0.249 | - | - | **0.69 (0.48–0.99)** | **0.043** | 1.04 (0.61–1.78) | 0.891 | 1.02 (0.67–1.53) | 0.935 | - | - |
| ≥100,000 | 0.54 (0.18–1.56) | 0.251 | - | - | 1.04 (0.54–2.00) | 0.909 | 1.89 (0.70–5.10) | 0.206 | 0.97 (0.44–2.13) | 0.947 | - | - |
| **Wealth tertiles** | | | | | | | | | | | | |
| Poorest | 1 | - | - | - | 1 | - | 1 | - | 1 | - | - | - |
| Middle | 0.74 (0.45–1.24) | 0.251 | - | - | 0.62 (0.42–0.90) | 0.011 | 0.59 (0.33–1.06) | 0.077 | 1.05 (0.65–1.69) | 0.845 | - | - |
| Richest | 0.76 (0.46–1.24) | 0.272 | - | - | **0.48 (0.32–0.71)** | **0.000** | **0.44 (0.23–0.82)** | **0.011** | 1.11 (0.70–1.76) | 0.655 | - | - |
| **Fetal characteristics** | | | | | | | | | | | | |
| **Gender** | | | | | | | | | | | | |
| Male | **1** | - | **1** | - | **1** | - | **1** | - | **1** | - | - | - |
| Female | **1.88 (1.21–2.93)** | **0.005** | **2.00 (1.13–3.52)** | **0.017** | **1.43 (1.03–2.00)** | **0.034** | 1.40 (0.88–2.24) | 0.159 | 0.89 (0.62–1.29) | 0.543 | - | - |
| **Parity** | | | | | | | | | | | | |
| Nullipara | 1 | - | - | - | 1 | - | - | - | 1 | - | 1 | - |
| 1–3 | 0.82 (0.54–1.24) | 0.342 | - | - | 1.25 (0.90–1.73) | 0.188 | - | - | **0.63 (0.44–0.91)** | **0.014** | 0.71 (0.35–1.45) | 0.345 |
| ≥4 | 1.07 (0.40–2.81) | 0.899 | - | - | 1.09 (0.50–2.40) | 0.826 | - | - | 0.52 (0.18–1.50) | 0.224 | 0.37 (0.08–1.83) | 0.222 |

*(Continued)*

**Table 4.** (Continued)

| Variables | Low Birth Weight | | | | Preterm Birth | | | | Birth Asphyxia | | | |
|---|---|---|---|---|---|---|---|---|---|---|---|---|
| Sociodemographic characteristics | Unadjusted | P-value | Adjusted | P-value | Unadjusted | P-value | Adjusted | P-value | Unadjusted | P-value | Adjusted | P-value |
| **Mode of delivery** | | | | | | | | | | | | |
| SVD | **0.56 (0.37–0.84)** | **0.005** | 1.02 (0.49–2.09) | 0.969 | **0.70 (0.51–0.96)** | **0.027** | 0.75 (0.42–1.34) | 0.337 | **0.42 (0.29–0.60)** | **0.000** | 1.40 (0.31–6.33) | 0.660 |
| Elective CS | 1.41 (0.84–2.36) | 0.197 | – | – | 1.14 (0.74–1.75) | 0.561 | – | – | **1.66 (1.07–2.57)** | **0.024** | 1.96 (0.40–9.69) | 0.409 |
| Emergency CS | **2.09 (1.32–3.29)** | **0.002** | **2.40 (1.06–5.42)** | **0.036** | **1.63 (1.12–2.38)** | **0.011** | 1.46 (0.73–2.93) | 0.291 | **2.25 (1.49–3.40)** | **0.000** | 2.46 (0.51–11.75) | 0.260 |
| **History of CS** | | | | | | | | | | | | |
| No | 1 | – | – | – | 1 | – | – | – | 1 | – | 1 | – |
| Yes | 1.03 (0.57–1.86) | 0.920 | – | – | 1.30 (0.84–2.00) | 0.230 | – | – | **2.02 (1.21–3.35)** | **0.007** | 1.90 (0.96–3.75) | 0.065 |
| **History of gestational HBP** | | | | | | | | | | | | |
| No | 1 | – | 1 | – | 1 | – | – | – | 1 | – | – | – |
| Yes | **3.00 (1.42–6.31)** | **0.004** | **3.34 (1.48–7.52)** | **0.004** | 1.70 (0.84–3.43) | 0.139 | – | – | 1.59 (0.62–4.04) | 0.326 | – | – |
| **History of stillbirths** | | | | | | | | | | | | |
| Yes | 1.83 (0.98–3.43) | 0.059 | – | – | **1.84 (1.13–2.99)** | **0.014** | **2.05 (1.14–3.68)** | **0.017** | 1.28 (0.69–2.35) | 0.438 | – | – |
| No | 1 | – | – | – | 1 | – | 1 | – | 1 | – | – | – |
| **History of miscarriage** | | | | | | | | | | | | |
| Yes | 1.29 (0.78–2.13) | 0.330 | – | – | 0.70 (0.46–1.06) | 0.088 | – | – | **1.71 (1.09–2.71)** | **0.021** | 1.49 (0.86–2.59) | 0.153 |
| No | 1 | – | – | – | 1 | – | – | – | 1 | – | 1 | – |
| **Chronic Medical Disease** | | | | | | | | | | | | |
| Yes | 1.22 (0.65–2.30) | 0.533 | – | – | 0.89 (0.52–1.52) | 0.669 | – | – | 1.26 (0.73–2.19) | 0.403 | – | – |
| No | 1 | – | – | – | 1 | – | – | – | 1 | – | – | – |
| **Maternal BMI** | | | | | | | | | | | | |
| Underweight | 1 | – | – | – | 1 | – | – | – | 1 | – | – | – |
| Normal weight | 0.86 (0.25–2.93) | 0.804 | – | – | 1.84 (0.55–6.16) | 0.322 | – | – | 0.88 (0.25–3.13) | 0.842 | – | – |
| Overweight | 0.71 (0.20–2.51) | 0.597 | – | – | 1.66 (0.49–5.65) | 0.417 | – | – | 0.95 (0.26–3.41) | 0.931 | – | – |
| Obese | 0.52 (0.14–1.95) | 0.330 | – | – | 1.64 (0.47–5.70) | 0.436 | – | – | 1.36 (0.37–4.99) | 0.640 | – | – |
| **Antenatal visits** | | | | | | | | | | | | |
| < 4 visits | 1 | – | – | – | 1 | – | – | – | 1 | – | – | – |
| ≥ 4 visits | 1.14 (0.58–2.22) | 0.709 | – | – | 0.89 (0.55–1.44) | 0.634 | – | – | 0.97 (0.57–1.65) | 0.904 | – | – |

**Table 5. Lifestyle and behavioural factors (physical activity, sedentary behaviors and dietary patterns) associated with adverse perinatal outcomes in Ibadan, Nigeria.**

| Variables | Low Birth Weight | | | | Preterm Birth | | | | Birth Asphyxia | | | |
|---|---|---|---|---|---|---|---|---|---|---|---|---|
| Lifestyles/behaviours characteristics | Unadjusted | P-value | Adjusted | P-value | Unadjusted | P-value | Adjusted | P-value | Unadjusted | P-value | Adjusted | P-value |
| **Alcohol use** | | | | | | | | | | | | |
| Yes | 0.84 (0.44–1.60) | 0.590 | – | – | 1.04 (0.65–1.66) | 0.874 | – | – | 0.76 (0.42–1.37) | 0.356 | – | – |
| No | 1 | – | – | – | 1 | – | – | – | 1 | – | – | – |
| **Tobacco use** | | | | | | | | | | | | |
| Yes | 0.55 (0.13–2.31) | 0.412 | – | – | 1.26 (0.58–2.75) | 0.564 | – | – | 1.52 (0.64–3.60) | 0.340 | – | – |
| No | 1 | – | – | – | 1 | – | – | – | 1 | – | – | – |
| **Perceived stress** | | | | | | | | | | | | |
| Low | 1 | – | – | – | 1 | – | – | – | 1 | – | 1 | – |
| Moderate | 0.63 (0.34–1.19) | 0.153 | – | – | 1.07 (0.60–1.89) | 0.831 | – | – | **0.54 (0.31–0.92)** | **0.024** | 1.07 (0.47–2.44) | 0.864 |
| High | 0.51 (0.16–1.63) | 0.256 | – | – | 1.01 (0.42–2.46) | 0.976 | – | – | 0.41 (0.15–1.10) | 0.076 | 0.55 (0.11–2.84) | 0.474 |
| **Depression** | | | | | | | | | | | | |
| Yes | 0.83 (0.45–1.53) | 0.547 | – | – | **1.73 (1.17–2.55)** | **0.006** | **1.87 (1.08–3.25)** | **0.027** | 1.07 (0.65–1.77) | 0.792 | – | – |
| No | 1 | – | – | – | 1 | – | 1 | – | 1 | – | – | – |
| **Unprescribed medicines** | | | | | | | | | | | | |
| Yes | 0.42 (0.17–1.03) | 0.059 | – | – | 0.91 (0.49–1.69) | 0.768 | – | – | 1.52 (0.86–2.67) | 0.148 | – | – |
| No | 1 | – | – | – | 1 | – | – | – | 1 | – | – | – |
| **Herbal** | | | | | | | | | | | | |
| Yes | 0.60 (0.23–1.58) | 0.300 | – | – | 1.51 (0.82–2.79) | 0.186 | – | – | 1.61 (0.85–3.02) | 0.143 | – | – |
| No | 1 | – | – | – | 1 | – | – | – | 1 | – | – | – |
| **Physical Activity & Sedentary Behaviours** | | | | | | | | | | | | |
| **Sedentary intensity activity** | | | | | | | | | | | | |
| Low | 1 | – | – | – | – | – | 1 | – | 1 | – | – | – |
| Medium | 1.52 (0.93–2.50) | 0.099 | – | – | **0.63 (0.43–0.93)** | **0.021** | 0.81 (0.46–1.43) | 0.472 | 0.95 (0.60–1.51) | 0.836 | – | – |
| High | 1.01 (0.60–1.72) | 0.959 | – | – | 0.74 (0.51–1.08) | 0.117 | 0.92 (0.52–1.64) | 0.776 | 1.12 (0.73–1.72) | 0.612 | – | – |
| **Light intensity activity** | | | | | | | | | | | | |
| Low | 1 | – | – | – | 1 | – | – | – | 1 | – | – | – |
| Medium | 0.81 (0.49–1.34) | 0.409 | – | – | 1.23 (0.84–1.80) | 0.287 | – | – | 0.64 (0.41–1.00) | 0.051 | – | – |
| High | 0.97 (0.60–1.58) | 0.908 | – | – | 1.00 (0.67–1.48) | 0.987 | – | – | 0.75 (0.49–1.16) | 0.196 | – | – |
| **Moderate intensity** | | | | | | | | | | | | |
| Low | 1 | – | – | – | 1 | – | 1 | – | 1 | – | – | – |
| Medium | 1.21 (0.75–1.97) | 0.437 | – | – | **1.48 (1.01–2.17)** | **0.043** | 1.25 (0.66–2.35) | 0.491 | 0.88 (0.57–1.36) | 0.566 | – | – |
| High | 0.92 (0.55–1.54) | 0.743 | – | – | 1.14 (0.76–1.70) | 0.532 | 0.75 (0.38–1.49) | 0.407 | 0.79 (0.51–1.23) | 0.302 | – | – |
| **Vigorous activity** | | | | | | | | | | | | |
| Low | 1 | – | – | – | 1 | – | – | – | 1 | – | – | – |
| Medium | 1.04 (0.58–1.84) | 0.901 | – | – | 1.03 (0.65–1.61) | 0.915 | – | – | 1.18 (0.71–1.96) | 0.524 | – | – |
| High | 0.93 (0.56–1.55) | 0.786 | – | – | 0.97 (0.65–1.43) | 0.860 | – | – | 1.34 (0.87–2.06) | 0.184 | – | – |
| **Occupation-related activity** | | | | | | | | | | | | |
| Low | 1 | – | – | – | 1 | – | – | – | 1 | – | – | – |

*(Continued)*

**Table 5.** (Continued)

| Variables | Low Birth Weight | | | | Preterm Birth | | | | Birth Asphyxia | | | |
|---|---|---|---|---|---|---|---|---|---|---|---|---|
| Lifestyles/behaviours characteristics | Unadjusted | P-value | Adjusted | P-value | Unadjusted | P-value | Adjusted | P-value | Unadjusted | P-value | Adjusted | P-value |
| Medium | 1.05 (0.65–1.69) | 0.845 | – | – | 0.84 (0.57–1.22) | 0.350 | – | – | 1.26 (0.83–1.92) | 0.282 | – | – |
| High | 0.83 (0.49–1.39) | 0.468 | – | – | 0.80 (0.54–1.17) | 0.252 | – | – | 0.72 (0.45–1.16) | 0.176 | – | – |
| **Transport-related activity** | | | | | | | | | | | | |
| Low | 1 | – | – | – | 1 | – | – | – | 1 | – | – | – |
| Medium | 1.09 (0.67–1.78) | 0.738 | – | – | 1.41 (0.96–2.08) | 0.081 | – | – | 0.84 (0.55–1.29) | 0.435 | – | – |
| High | 0.98 (0.60–1.61) | 0.931 | – | – | 1.41 (0.96–2.07) | 0.084 | – | – | 0.83 (0.53–1.28) | 0.388 | – | – |
| **Household/caregiving activity** | | | | | | | | | | | | |
| Low | 1 | – | – | – | 1 | – | 1 | – | 1 | – | – | – |
| Medium | 0.99 (0.59–1.66) | 0.970 | – | – | 1.46 (0.98–2.18) | 0.063 | 1.27 (0.65–2.48) | 0.489 | 0.88 (0.56–1.37) | 0.569 | – | – |
| High | 1.22 (0.75–2.00) | 0.428 | – | – | **1.53 (1.03–2.27)** | **0.035** | 1.56 (0.79–3.07) | 0.202 | 0.91 (0.59–1.39) | 0.658 | – | – |
| **Sports activity** | | | | | | | | | | | | |
| Low | 1 | – | – | – | 1 | – | – | – | 1 | – | – | – |
| Medium | 1.05 (0.63–1.75) | 0.860 | – | – | 1.13 (0.77–1.68) | 0.531 | – | – | 1.19 (0.77–1.86) | 0.439 | – | – |
| High | 1.31 (0.80–2.14) | 0.279 | – | – | 1.37 (0.93–2.00) | 0.109 | – | – | 1.23 (0.79–1.91) | 0.363 | – | – |
| **Dietary patterns** | | | | | | | | | | | | |
| **Protein-rich diet and non-alcoholic beverages** | | | | | | | | | | | | |
| Low | 1 | – | – | – | 1 | – | 1 | – | 1 | – | – | – |
| Medium | 1.39 (0.87–2.23) | 0.175 | – | – | 0.88 (0.61–1.26) | 0.488 | 0.68 (0.39–1.17) | 0.161 | 0.91 (0.59–1.41) | 0.669 | – | – |
| High | 0.77 (0.45–1.33) | 0.348 | – | – | **0.52 (0.34–0.78)** | **0.002** | **0.50 (0.28–0.89)** | **0.019** | 1.01 (0.65–1.57) | 0.967 | – | – |
| **Fruits Diet** | | | | | | | | | | | | |
| Low | 1 | – | – | – | 1 | – | – | – | 1 | – | – | – |
| Medium | 1.16 (0.70–1.93) | 0.558 | – | – | 1.38 (0.95–1.99) | 0.088 | – | – | 1.37 (0.88–2.13) | 0.161 | – | – |
| High | 1.24 (0.75–2.04) | 0.408 | – | – | 0.76 (0.50–1.15) | 0.194 | – | – | 1.23 (0.79–1.94) | 0.362 | – | – |
| **Typical diet with alcohol** | | | | | | | | | | | | |
| Low | 1 | – | – | – | 1 | – | – | – | 1 | – | – | – |
| Medium | 1.08 (0.66–1.77) | 0.767 | – | – | 1.35 (0.93–1.98) | 0.117 | – | – | 1.16 (0.75–1.79) | 0.510 | – | – |
| High | 1.06 (0.64–1.75) | 0.823 | – | – | 0.99 (0.67–1.48) | 0.971 | – | – | 1.04 (0.67–1.63) | 0.849 | – | – |
| **Legumes** | | | | | | | | | | | | |
| Low | [21]1 | – | – | – | 1 | – | – | – | 1 | – | – | – |
| Medium | 0.70 (0.42–1.16) | 0.168 | – | – | 1.31 (0.90–1.90) | 0.155 | – | – | 0.71 (0.46–1.11) | 0.131 | – | – |
| High | 0.96 (0.59–1.55) | 0.863 | – | – | 0.99 (0.66–1.49) | 0.970 | – | – | 1.05 (0.68–1.61) | 0.842 | – | – |
| **Refined grains** | | | | | | | | | | | | |
| Low | 1 | – | – | – | 1 | – | – | – | 1 | – | – | – |
| Medium | 1.20 (0.73–1.97) | 0.480 | – | – | 1.28 (0.88–1.86) | 0.202 | – | – | 0.92 (0.59–1.43) | 0.709 | – | – |
| High | 1.05 (0.63–1.76) | 0.843 | – | – | 0.89 (0.59–1.33) | 0.557 | – | – | 0.96 (0.62–1.49) | 0.852 | – | – |

p=0.006), being born female (UOR=1.43, 95% CI: (1.03–2.00) p=0.034), and a high protein diet (UOR=0.52, 95% CI: (0.34–0.78) p=0.002). Physical activity had a significant association with PTB in the unadjusted model: medium level of moderate-intensity physical activity [UOR=1.5;95% CI: (1.01–2.17); p-value=0.043], high levels of household/caregiving activity; [UOR=1.5; 95% CI:(1.03–2.27); p-value=0.035] had positive associations with PTB while medium level of sedentary-intensity physical activity [UOR=0.6; 95% CI:(0.43–0.93); p-value<0.021] was inverse. The associations became insignificant after adjustment.

The significant factors in the adjusted models were women in the richest tertile [AOR=0.44; 95% CI:(0.23–0.82); p-value=0.011]. A history of stillbirth [AOR=2.05; 95%CI:(1.14–3.68); p-value=0.017]; antepartum depression [AOR=1.73; 95% CI:(1.08–3.25); p-value=0.027]. A high-protein diet [AOR=0.50;, 95% CI: (0.34–0.73); p-value=0.019].

### Prevalence and associated factors of birth asphyxia

The prevalence of birth asphyxia was 16.3%. The factors associated with birth asphyxia in the unadjusted model were high parity [UOR=0.6; 95% CI:(0.44–0.91); p-value<0.014].; mode of delivery: spontaneous vertex delivery: [UOR=0.4; 95% CI:(0.29–0.60); p-value<0.001, elective caesarean section [UOR=1.7; 95% CI:(1.07–2.57); p-value=0.024], emergency caesarean section, [UOR=2.3; 95% CI:(1.49–3.40); p-value<0.001]; histories of miscarriage [UOR=1.7; 95% CI:(1.09–2.71); p-value=0.021], and caesarean section [UOR=2.0; 95% CI:(1.21–3.35); p-value<0.007], and moderate stress [UOR=0.5; 95% CI:(0.31–0.92); p-value=0.024]. However, none were found to be significantly linked to birth asphyxia after adjusting for other variables.

## Discussion

Adverse perinatal outcomes contribute significantly to the burden of high neonatal and infant morbidity and mortality in LMICs, including Nigeria. Notably, SDG 3.2 seeks to end preventable deaths of newborns and under-five children to reduce neonatal mortality (NNM) to at least 12 per 1000 live births globally [30]. Meanwhile, the current NNM in Nigeria is 35 per 1000 live births [2]. Maternal health and perinatal outcomes are complexly linked with maternal complications, often increasing the risk of APOSs [31]. Hence, we posited that maternal lifestyle would likely mitigate APO, an association rarely explored in Nigeria. Therefore, we assessed APO's determinants, especially the influence of maternal lifestyle. We also examined the broad range of determinants associated with specific outcomes (LBW, premature birth and birth asphyxia). The maternal lifestyle characteristics examined included dietary patterns, physical activity, alcohol use, tobacco exposure and drug use in pregnancy. Notably, studies on APOS as a construct are sparse in Nigeria; instead, researchers have assessed the individual components such as LBW, PTB and birth asphyxia. Overall, the prevalence of APOS was 26.7% in our study, which is lower than the prevalence of 29.7% reported by Tamirat *et al.* (2023) that examined APOs in ten Sub-Saharan countries using data from the Demographic Health Surveys [15] and a prevalence of 31.5% in South West Ethiopia [32]. However, this is higher compared with other reports from Northwest Ethiopia (19.4%) [22], South Ethiopia (18.6%) [25], and Tigray Ethiopia (22.5%) [26]. The differences in these studies could be attributed to the study area, study population, study design and the data collection method.

Importantly, the factors associated with APO among our study population after adjusting for confounding variables included occupationally related physical activity, a history of stillbirth, and operative delivery. Even though the physical activity level among Nigerian pregnant women is reportedly very low, we found that women who engaged in higher tertiles of occupationally related physical activity had 31% lower odds of APO than women with lower physical activity. Physical activity may provide this protective effect by reducing the risk

of pregnancy complications such as preeclampsia, maternal obesity and gestational diabetes which increase the risk of APO [33–36].

This finding lends credence to the deliberate role of physical activity as part of maternal and perinatal health interventions and services. For example, some studies have reported the protective effect PA and certain pregnancy complications. However, more studies need to emanate from Africa were studies are currently lacking compared with the Caucasian and Asian populations. Additionally safety concerns sometimes prevent pregnant women from optimizing PA because it thought to reduce placental circulation by shunting blood to the skeletal muscles, and increasing catecholamines that stimulates uterine contractility [37]. Notably, Rego *et al.*, 2016 in establishing no relationship between women's level of physical activity and any adverse perinatal outcomes (IUGR, PTD LBW), among Brazilian pregnant women [38] supported the safety of physical activity in not increasing the risk of fetal loss or prematurity.

The determinants of LBW in this study were the history of gestational hypertension, delivery by emergency CS and being a female newborn. Importantly, we found that women with gestational hypertension had a threefold odd (AOR=3.34) of having LBW babies. The relationship between hypertension during pregnancy and LBW are well reported in the literature [39–42] a systematic review and meta-analysis that examined the influence of pregnancy-induced hypertension on LBW reported a fourfold odd of LBW among women with PIH compared with normotensive women [39]]. The biological mechanism has been attributed to the incomplete trophoblast invasion into spiral arteries in the placenta leading to reduced uteroplacental blood perfusion, intrauterine growth retardation and LBW [43,44]. The provision of adequate care is essential for women with a history of PIH, which includes calcium supplementation, low-dose aspirin, prenatal care and timely delivery. The emergency caesarean section was associated with LBW by twofold in our study. Other researchers have confirmed the association [45,46]. For example, Pires-Menard *et al.* (2021) reported that birth by emergency CS had a higher risk of adverse birth outcomes [45]. Generally, emergency CS is usually indicated by maternal-fetal complications and is often performed to save the life of the mother, fetus or both. The cesarean section also protects low birth weight babies by preventing potential compressions at birth [47], thereby increasing newborn survival rates [46]. Even though lifestyle modifications during pregnancy could provide cost-effective and preventive interventions, this has not received much attention in sub-Saharan Africa. Nonetheless, a Zambia study in 2021 reported passive tobacco smoking during pregnancy, alcohol intake during pregnancy and inadequate dietary intake during pregnancy were associated with LBW [48,49]. However, this study found no significant association between lifestyle factors and LBW.

Notably, the factors associated with PTB were household wealth, history of stillbirth, antepartum depression, and high protein diet. The relationship with physical activity became insignificant after adjusting for confounders. We found that rich women had lower odds of having PTB. Women with wealth and high income are furnished with the resources for a healthy lifestyle and maternal dietary patterns. This promotes optimal growth and development and lowers risks of pregnancy complications and adverse pregnancy outcomes [50–53]. Olatona *et al.* (2021) reported that pregnant women with higher socio-economic status reported healthy dietary habits and higher dietary diversity scores [54] Notably, of the five dietary patterns examined in this study [25], only the high protein diet and non-alcoholic beverage dietary pattern, which explained the highest variance, was protective of PTB (AOR=0.5). The protein diet and non-alcoholic beverage dietary pattern were characterised by a high adherence to red meat, fish, eggs, green vegetables, cream milk and soft drinks, cocoa beverages, and pastries [25] This dietary pattern was high

in protein and contained anti-inflammatory (e.g., fish, eggs, green vegetables) and pro-inflammatory foods (soft drinks and pastries).

Interestingly, researchers have begun examining maternal dietary patterns' associations with inflammatory potential and APO in Europe [55,56] USA [57,58], Spain [59], Iran [60], Japan [61] and China [62]. These studies hinge on the Developmental Origins of Health, which hypothesizes that pre-pregnancy and intrauterine nutrition can influence pregnancy outcomes and the infants' future health. In our study, the high-protein diet and non-alcoholic beverage dietary pattern reduced the odds of PTB by twofold because of their high anti-inflammatory food content. Other researchers have confirmed this finding [62–65], even though these associations are yet to be explored in sub-Saharan Africa, including Nigeria. Comparably, the Japan Environment and Children's Study, Ishibashi *et al.* (2019) [66] found that a pro-inflammatory diet increased the risk of PTB and LBW. Although the mechanism of the association between maternal pro-inflammatory diet and PTB is not fully understood, it is thought to be due to systemic inflammation caused by the release of pro-inflammatory cytokines, which produce prostaglandins (stimulates uterine contraction) and matrix-degrading enzymes (preterm rupture of membranes) both increase the risk of PTB. Systemic inflammation may also be associated with co-morbidities such as hypertension that may indicate medically induced PTB [55,66]. Our finding is essential for the formulation of dietary guidelines during antenatal care.

In this study, physical activity had no significantly association with PTB on in the adjusted model. Notably, scientists have reported conflicting results on the association between physical activity and PTB; some have reported no association [56,57,67], and a few have reported a positive association [68] and inverse association [69].

Further research is needed to examine the association between physical activity and PTB, especially in sub-Saharan Africa, where little attention has been given to the influence of maternal lifestyle on perinatal outcomes. However, the physiological pathways by which physical activity prevents preterm birth may include improving maternal mental health, including mood and reducing depression during pregnancy, increasing insulin sensitivity, reducing inflammatory responses, and reducing oxidative stress [70,71]. For instance, we found that women with antepartum depression in our study had a significantly higher risk of PTB in our study population [17]. Additionally, women with a history of stillbirth had an increased risk of PTB, indicated by underlying factors and co-morbidities, which results in maternal complications and recurring adverse birth outcomes [72–74]]. The finding emphasizes the need for proper evaluation of pregnant women during antenatal care, including their past obstetric history for preventing PTB and other APO.

This research found a positive association with a history of miscarriage in a previous pregnancy and caesarean section (EMCS & ELCS) and a negative association with vagina delivery mode of delivery, primiparous mothers, moderate stress level, and history of CS. In this study, nulliparous women had a higher risk for birth asphyxia, an association confirmed by other studies [75–77]. For example, a recent study from Pakistan reported a higher prevalence of birth asphyxia among nulliparous women due to a lack of knowledge and poor birth preparedness, which may increase the risk of pregnancy complications [78]. Additionally, operational deliveries were associated with birth asphyxia. In contrast, spontaneous vaginal delivery had a negative association in the study, as reported by others, due to complications during labour, including prolonged labour, fetal distress, delays in decision-making and access among mothers with complications, particularly obtaining emergency obstetric care [75,79,80]. However, after adjusting for confounders, none were found to be significantly related to Birth Asphyxia. Importantly, none of the lifestyle characteristics examined had a significant statistical association with birth asphyxia.

Our study contributes to maternal and perinatal epidemiology in Nigeria, particularly by examining the influence of maternal lifestyle on APO in Nigeria using the Ibadan Pregnancy Cohort Study data. We also discussed a broad range of lifestyle factors associated with APO in Nigeria compared with previous studies which focused on maternal nutrition and cigarette smoking. Conversely, it has some limitations, mainly loss to follow-up bias and limited external validity to women who had obtained care from primary healthcare settings or rural areas. Using self-reported questionnaires to assess maternal lifestyle may be associated with recall and misclassification bias from under or over-reporting. Importantly, maternal lifestyle was assessed just once early in pregnancy and not throughout pregnancy. Hence, future research needs to explore these associations using objective measures, e.g., pedometers or accelerometers and assessing maternal lifestyle at multiple times.

## Conclusion and implication of the study

The occurrence of adverse perinatal outcomes was high among our study population, occurring in one in every four pregnancies. The maternal lifestyle factors associated with APO in our study population were – occupationally related PA, protein rich diet and non-alcoholic beverages and antepartum depression. The other determinants were social demographic factors and poor obstetric history. Therefore, adequate maternal dietary pattern physical activity and providing maternal mental health support, which are modifiable factors and cost-effective interventions, should be targeted against APOS in Nigeria's health care system. Healthcare professionals can focus on promoting healthy lifestyle choices, providing mental health support, and encouraging quality antenatal care to reduce the risk of adverse perinatal outcomes.

## Supporting information

**S1 Text**. Supporting Information.
(DOCX)

## Acknowledgments

We appreciate our research team – research nurses, laboratory scientists, numerous research assistants, and data personnel- for their dedication, support, and hard work. We also appreciate the health workers –doctors, nurses, clinic and record staff of the various health facilities for their cooperation and support in the four facilities: University College Hospital, Adeoyo Maternity Teaching Hospital, Jericho Specialist Hospital, and Saint Mary Catholic Hospital Oluyoro, Ibadan. We appreciate the input of CARTA (Consortium for Advanced Research Training for Africa) in all its training, care, support, oversight, funding, and sponsorship efforts

## Author contributions

**Conceptualization:** Ikeola A. Adeoye.

**Data curation:** Ikeola A. Adeoye.

**Formal analysis:** Ikeola A. Adeoye, Chioma O Unogu.

**Funding acquisition:** Ikeola A. Adeoye.

**Investigation:** Ikeola A. Adeoye, Chioma O Unogu.

**Methodology:** Ikeola A. Adeoye.

**Project administration:** Ikeola A. Adeoye.

**Resources:** Ikeola A. Adeoye.

**Software:** Ikeola A. Adeoye.

**Supervision:** Ikeola A. Adeoye, Kofoworola Adediran, Babatunde M Gbadebo.

**Validation:** Ikeola A. Adeoye, Chioma O Unogu, Kofoworola Adediran, Babatunde M Gbadebo.

**Visualization:** Ikeola A. Adeoye, Chioma O Unogu, Kofoworola Adediran, Babatunde M Gbadebo.

**Writing – original draft:** Chioma O Unogu.

**Writing – review & editing:** Ikeola A. Adeoye, Chioma O Unogu, Kofoworola Adediran, Babatunde M Gbadebo.

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
