## [Decision Letter · Decision Letter 0]

9 May 2024

PGPH-D-24-00167

DETERMINANTS OF ADVERSE PERINATAL OUTCOMES IN IBADAN, NIGERIA: THE INFLUENCE OF MATERNAL LIFESTYLE

Dear Dr. Adeoye,

Thank you for submitting your manuscript to PLOS Global Public Health. After careful consideration, we feel that it has merit but does not fully meet PLOS Global Public Health’s publication criteria as it currently stands. Therefore, we invite you to submit a revised version of the manuscript that addresses the points raised during the review process.

Please note that we have only been able to secure a single reviewer to assess your manuscript. We are issuing a decision on your manuscript at this point to prevent further delays in the evaluation of your manuscript. Please be aware that the editor who handles your revised manuscript might find it necessary to invite additional reviewers to assess this work once the revised manuscript is submitted. However, we will aim to proceed on the basis of this single review if possible. 

Please see the attached document for the reviewer's comments. Could you please revise the manuscript to carefully address the concerns raised?

We look forward to receiving your revised manuscript.

Kind regards,

Steve Zimmerman, PhD

PLOS Staff Editor

Journal Requirements:

Additional Editor Comments (if provided):

Reviewers' comments:

Reviewer's Responses to Questions

**Comments to the Author**

1. Does this manuscript meet PLOS Global Public Health’s publication criteria ? Is the manuscript technically sound, and do the data support the conclusions? The manuscript must describe methodologically and ethically rigorous research with conclusions that are appropriately drawn based on the data presented.

Reviewer #1: Yes

2. Has the statistical analysis been performed appropriately and rigorously?

Reviewer #1: No

3. Have the authors made all data underlying the findings in their manuscript fully available (please refer to the Data Availability Statement at the start of the manuscript PDF file)?

Reviewer #1: Yes

4. Is the manuscript presented in an intelligible fashion and written in standard English?

Reviewer #1: No

5. Review Comments to the Author

Reviewer #1: This paper refers to a cohort of pregnant women with the analysis of lifestyle status and adverse perinatal outcomes in Nigeria. I appreciate the editorial board's invitation to review this paper and congratulate the authors for the study. Please find some comments in the file uploaded.

6. PLOS authors have the option to publish the peer review history of their article (what does this mean? ). If published, this will include your full peer review and any attached files.

**Do you want your identity to be public for this peer review?** For information about this choice, including consent withdrawal, please see our Privacy Policy .

Reviewer #1: No

While revising your submission, please upload your figure files to the Preflight Analysis and Conversion Engine (PACE) digital diagnostic tool, https://pacev2.apexcovantage.com/ . PACE helps ensure that figures meet PLOS requirements. To use PACE, you must first register as a user. Registration is free. Then, login and navigate to the UPLOAD tab, where you will find detailed instructions on how to use the tool. If you encounter any issues or have any questions when using PACE, please email PLOS at figures@plos.org. Please note that Supporting Information files do not need this step.

---

## [Editor Report · Decision Letter 1]

21 Oct 2024

PGPH-D-24-00167R1

Determinants of Adverse Perinatal Outcomes in Ibadan, Nigeria: The influence of maternal lifestyle

Dear Dr. Adeoye,

Thank you for submitting your manuscript to PLOS Global Public Health. After careful consideration, we feel that it has merit but does not fully meet PLOS Global Public Health’s publication criteria as it currently stands. Therefore, we invite you to submit a revised version of the manuscript that addresses the points raised during the review process.

We look forward to receiving your revised manuscript.

Kind regards,

Mohammad Shahidul Islam, PhD

Academic Editor

Journal Requirements:

Additional Editor Comments (if provided):

In their study, Adeoye et al. reported the determinants of adverse pregnancy outcomes (APOs) in Nigeria and their association with maternal lifestyle. The data presented are important and timely, as APOs significantly impact perinatal and neonatal survival, increasing the risk of child mortality, developmental disabilities, and lifelong ill health. Countries in sub-Saharan Africa bear a disproportionate share of the global burden of APOs, yet few studies from the region have reported on the factors associated with these outcomes. However, the authors should consider addressing the following issues to further enhance the value of the paper.

1. The abstract is lengthy; please consider condensing it to enhance readability.

2. In the methods section, the analysis plan needs to be more explicit in explaining how the multivariate logistic regression model was built to calculate the adjusted odds ratios.

3. Table 1 and elsewhere indicate that percentages were calculated using a denominator of 1,339, which does not seem appropriate, as Figure 1 shows that newborn outcomes could not be tracked for 60 deliveries. Please correct the denominators or clarify why this denominator is being used.

4. Please include appropriate statistical test values in Tables 1 and 2 to indicate whether there are any significant differences in the prevalence of adverse pregnancy outcomes (APOs) between the sub-groups of mothers or neonates.

5. In Tables 1 and 2, please indicate in the footnotes if more than 5% of data are missing for any particular variable.

6. The paper would flow better if the findings related to lifestyle determinants are presented first, as this is the primary objective of the manuscript.

7. In Tables 3 and 4, please add the number of events for each category, as knowing the number of APOs for each category is essential to understanding the significance of the data.

8. While the data on each adverse outcome in Tables 3 and 4 are valuable, it would have greater policy impact if the effect of each determinant were calculated to assess its overall impact on APOs.

9. In line 526, it states that physical activities increase the odds of preterm birth (PTB), but the data do not support this finding. This could mislead the reader, as some previous reports suggest that moderate physical work may reduce adverse pregnancy outcomes (APOs).

10. In lines 410-412, presenting only p-values is insufficient, as the determinants can affect the outcome in either direction. Please include the associated odds ratios (OR) for each variable.

11. Please highlight some limitations of the paper that may affect the outcomes of the findings.

12. In the conclusion, the authors emphasized promoting a healthy lifestyle without specifying which aspects of lifestyle should be promoted. Please clarify those.

---

## [Editor Report · Decision Letter 2]

3 Jan 2025

Determinants of Adverse Perinatal Outcomes in Ibadan, Nigeria: The influence of maternal lifestyle

PGPH-D-24-00167R2

Dear Dr. Adeoye,

We are pleased to inform you that your manuscript 'Determinants of Adverse Perinatal Outcomes in Ibadan, Nigeria: The influence of maternal lifestyle' has been provisionally accepted for publication in PLOS Global Public Health.

Best regards,

Mohammad Shahidul Islam, PhD

Academic Editor
